# Donor Splice Site Variant in *SLC9A6* Causes Christianson Syndrome in a Lithuanian Family: A Case Report

**DOI:** 10.3390/medicina58030351

**Published:** 2022-02-26

**Authors:** Gunda Petraitytė, Violeta Mikštienė, Evelina Siavrienė, Loreta Cimbalistienė, Živilė Maldžienė, Tautvydas Rančelis, Evelina Marija Vaitėnienė, Laima Ambrozaitytė, Justas Dapkūnas, Ramūnas Dzindzalieta, Erinija Pranckevičienė, Vaidutis Kučinskas, Algirdas Utkus, Eglė Preikšaitienė

**Affiliations:** 1Department of Human and Medical Genetics, Institute of Biomedical Sciences, Faculty of Medicine, Vilnius University, LT-08661 Vilnius, Lithuania; evelina.siavriene@mf.vu.lt (E.S.); loreta.cimbalistiene@mf.vu.lt (L.C.); zivile.maldziene@mf.vu.lt (Ž.M.); tautvydas.rancelis@mf.vu.lt (T.R.); evelina.vaiteniene@mf.vu.lt (E.M.V.); laima.ambrozaityte@mf.vu.lt (L.A.); erinija.pranckeviciene@mf.vu.lt (E.P.); vaidutis.kucinskas@mf.vu.lt (V.K.); algirdas.utkus@mf.vu.lt (A.U.); egle.preiksaitiene@mf.vu.lt (E.P.); 2Biobank of the Lithuanian Population and Rare Disorders, Institute of Biomedical Sciences, Faculty of Medicine, Vilnius University, LT-03101 Vilnius, Lithuania; violeta.mikstiene@mf.vu.lt (V.M.); ramunas.dzindzalieta@ittc.vu.lt (R.D.); 3Department of Bioinformatics, Institute of Biotechnology, Life Sciences Center, Vilnius University, LT-10257 Vilnius, Lithuania; justas.dapkunas@bti.vu.lt; 4Department of Software Engineering, Institute of Computer Science, Faculty of Mathematics and Informatics, Vilnius University, LT-03225 Vilnius, Lithuania

**Keywords:** *SLC9A6*, donor splice site variant, Christianson syndrome, cDNA analysis, protein modelling

## Abstract

*Background and Objectives*: The pathogenic variants of *SLC9A6* are a known cause of a rare, X-linked neurological disorder called Christianson syndrome (CS). The main characteristics of CS are developmental delay, intellectual disability, and neurological findings. This study investigated the genetic basis and explored the molecular changes that led to CS in two male siblings presenting with intellectual disability, epilepsy, behavioural problems, gastrointestinal dysfunction, poor height, and weight gain. *Materials and Methods*: Next-generation sequencing of a tetrad was applied to identify the DNA changes and Sanger sequencing of proband’s cDNA was used to evaluate the impact of a splice site variant on mRNA structure. Bioinformatical tools were used to investigate SLC9A6 protein structure changes. *Results*: Sequencing and bioinformatical analysis revealed a novel donor splice site variant (NC_000023.11(NM_001042537.1):c.899 + 1G > A) that leads to a frameshift and a premature stop codon. Protein structure modelling showed that the truncated protein is unlikely to form any functionally relevant SLC9A6 dimers. *Conclusions*: Molecular and bioinformatical analysis revealed the impact of a novel donor splice site variant in the *SLC9A6* gene that leads to truncated and functionally disrupted protein causing the phenotype of CS in the affected individuals.

## 1. Introduction

Christianson syndrome (CS; MIM# 300243) is an X-linked neurodevelopmental and regressive intellectual disability disorder that is caused by hemizygous pathogenic alterations in the *SLC9A6* gene (alternative names *NHE6* and *KIAA0267*; MIM# 300231). The phenotype of CS overlaps somewhat with Angelman syndrome (MIM# 105830) and was previously referred to as X-linked Angelman syndrome. However, possible developmental regression, progressive cerebellar atrophy, electrical status epilepticus during sleep, and shorter life span are differential characteristics specific to CS. Common clinical findings are moderate to severe intellectual disability, ataxia, early-onset drug resistant seizures, postnatal microcephaly, hypotonia, mutism, autistic behaviour, and generally happy demeanour [1,2,3]. The patients tend to develop dystonic movement and limb spasticity over time [3], and most affected individuals also have at least one of the following symptoms: abnormal eye movements, poor weight and height gain, gastroesophageal reflux, and high pain threshold [1,3,4]. Brain MRI frequently shows hippocampal or progressive cerebellar atrophy [2,4].

Female carriers exhibit a wide phenotypic spectrum from none to several phenotypic features because of the natural X inactivation process resulting in heterogenous cell populations. The commonly reported features of females are learning difficulties, speech, and language delay, mild to moderate intellectual disability, and behavioural issues, as well as mood and anxiety disorders more rarely [5]. M. F. Pescosolido et al. (2019) analysed 20 different pedigree carriers of *SLC9A6* variants and concluded that neuropsychological manifestations, such as deficits in attention and visuospatial function, were particularly frequent [6].

The protein encoded by *SLC9A6* belongs to the solute carrier (SLC) superfamily, which plays a major role in the transport of endogenous and exogenous compounds and has over 400 proteins classified into 65 subfamilies [7]. Specifically, the SLC9 gene family encodes Na^+^/H^+^ exchangers (NHEs), which are transmembrane proteins responsible for ion transport across lipid bilayers [8]. SLC9 genes are further divided into three subgroups: A, B, and C. The SLC9A subgroup includes nine isoforms (NHE1–9) [9]. *SLC9A6* codes the organellar (specifically endosomal) isoform SLC9A6, which is highly expressed in the central nervous system, explaining the neurological phenotype of CS patients. While detectable in all the nervous systems, analysis of *SLC9A6* knockout mice revealed the amygdala, hippocampus, and cerebral cortex as the regions most affected. Notably, *SLC9A6* expression in the brain is highest at the prenatal stage, lower at the postnatal stage, and significantly increased in adulthood [10]. *SLC9A6* is also found in zones of mineralising osteoblasts and various mitochondria-rich tissues, including the heart, skeletal muscles, liver, and pancreas, although the effect on these tissues is not yet fully understood [11,12].

In this study, we report the clinical findings of two brothers diagnosed with CS and their carrier mother. We also present bioinformatical and molecular analyses of a novel donor splice site variant in the *SLC9A6* gene.

## 2. Materials and Methods

### 2.1. Clinical Report

We report on three affected individuals in two generations of the family: two boys, aged 13 years and 9 years, and their mother (Figure 1A). The siblings have intellectual disability, epilepsy, severe global developmental delay, behavioural problems, gastrointestinal dysfunction, as well as poor height and weight gain. There is one unaffected sibling of the affected boys, a 12-year-old male. The patients’ uncle (mother’s brother) presented with similar clinical findings and died at the age of 9 years, but his medical records are not available. The patients’ father is healthy.

Patient 1: This patient is a 13-year-old male, the first child of non-consanguineous parents. He was born at 41 gestational weeks by caesarean delivery because his mother had an aortic aneurism. His birth weight was 3250 g (10–25th centile), his length was 58 cm (97th centile), and his head circumference 32 cm (<3rd centile), Apgar scores at 1 and 5 min were 9 and 9. Psychomotor developmental delay was noticed at 8 months of age together with poor weight gain and feeding difficulties. At the age of 11 months, he had his first episode of seizures. Later the seizures repeated about two times a month. A head CT showed signs of partial dysplasia of the temporal lobe, a sleep EEG showed irregular delta, alpha and theta wave polymorphic activity during sleep, and an ECG revealed a partial block of the right bundle branch. The patient has digestive problems (indigestion and vomiting) and frequent abdominal bloating, but no pathology was found after an examination by a gastroenterologist. An abdominal ultrasound was normal, and no visual or hearing impairment was diagnosed. He started walking at the age of 4.5 years. During his last examination at 12 years of age, his weight was 23 kg (<3rd centile), his height was 133 cm (<3rd centile), and his head circumference was 49 cm (<3rd centile), indicating microcephaly. He was able to speak only a few simple words. Stereotypical hand movements, hyper salivation, unstable gait, and inability to focus was observed.

Patient 2: This patient is a 9-year-old male, the third child in the family. He was born at 37 weeks of gestation via caesarean delivery following a normal pregnancy. Apgar scores at 1 and 5 min were 9 and 9. One and a half hours after birth, he had his first episode of generalised tonic-clonic seizures with respiratory failure. Seizures were refractory to treatment and repeated several times a day. An EEG registered generalised epileptic activity. Psychomotor developmental delay was noticed from infancy. He started walking with help at the age of 4 years. A brain MRI was normal. Ophthalmologic examination revealed hypermetropia of both eyes. He suffers from frequent constipation and eats only grated food. At age of 5 years, his weight was 15 kg (<3rd centile), his height was 99 cm (<3rd centile), and his head circumference was 47 cm (<3rd centile), indicating microcephaly. Minimal eye contact, no interest in toys, constant smiling, and putting fingers in his mouth was observed during examination.

Patient 3: The mother of the affected siblings, who is 40-years-old, has been diagnosed with a congenital heart defect and aortic aneurysm. She started walking and talking at the age of 5 years. She has a middle school education. Her height is 157 cm, her weight is 83 kg, and the circumference of her head is 54.5 cm.

### 2.2. Whole Exome Sequencing (WES)

Genomic DNA (gDNA) from probands and their parents was isolated from peripheral blood leukocytes using phenol–chloroform–isoamyl alcohol extraction method.

Whole exome sequencing using next-generation Applied Biosystems 5500 SOLiD™ platform after in-solution capture enrichment was used to sequence the samples. Sequencing protocol was carried out according to the optimized manufacturer’s protocols (Thermo Fisher Scientific, Life Technologies, South San Francisco, CA, USA) using Agilent SureSelectXT Target Enrichment System pooling six samples on one Flowchip. For Fragment Library preparation, 2 ng of genomic DNA was used. The shearing was performed by a Covaris S220 system. An average of 30× coverage was achieved.

Sequenced data were aligned to hg19 reference genome. The NGS pipeline (data alignment, quality control, variant calling) was performed using LifeScope™ Genomic Analysis Software v2.5.1. The annotation of NGS data was performed using the ANNOVAR v.2018Apr16 program [13]. American College of Human Genetics and Genomics criteria [14], in silico tools, and databases, which were provided by ANNOVAR program (SIFT, Polyphen2, GERP++, CADD, ExAC, GnomAD, 1000 Genome Project data, NCBI dbSNP, NCBI ClinVar) and the relevant scientific literature were used to assess the pathogenicity of the variants. Candidate genome variants were checked using the Integrative Genomics Viewer (IGV) visualization tool and validated using Sanger sequencing.

### 2.3. RNA Extraction and Complementary DNA Synthesis

Total RNA was extracted from the peripheral venous blood sample of proband 1 using Tempus™ Blood RNA Tube and Tempus™ Spin RNA Isolation Kit (Thermo Fisher Scientific, Austin, TX, USA) according to the optimized manufacturers’ protocols. Complementary DNA (cDNA) was synthesized from total RNA using a High-Capacity RNA-to-cDNA Kit (Thermo Fisher Scientific, Vilnius, Lithuania) following manufacturer’s protocol. Reverse transcription reactions were performed in a ProFlex PCR system (Thermo Fisher Scientific, Singapore) under the recommended conditions.

### 2.4. PCR

Amplification using specific primers for the fragments of *SLC9A6* gene was performed. PCR primers were designed using the Primer-BLAST (https://www.ncbi.nlm.nih.gov/tools/primer-blast/, accessed on 10 December 2020) tool. Primers for the analysis of gDNA samples were designed to amplify a region spanning a 3′ end of exon 6 and 5′ end of intron 6, while PCR amplifications of the cDNA sequence were performed using a forward primer designed on exon 5 and a reverse primer designed on exon 8 (Figure 1B). PCR products were fractioned by agarose gel electrophoresis and visualized under UV light. The images were acquired using a gel documentation system with a transilluminator (E.A.S.Y. 442K, Herolab, Germany) and analysed via E.A.S.Y. Win 32 image analysis software (Herolab, Germany).

### 2.5. Sanger Sequencing

DNA samples from the affected probands, their healthy brother, and mother, were analysed by Sanger sequencing for family segregation analysis. To elucidate the pathogenicity of the detected donor splice site variant, analysis by cDNA Sanger sequencing was performed for proband 1 who has the same clinical features as proband 2. PCR products were sequenced with the BigDye^®^ Terminator v3.1 Cycle Sequencing Kit (Thermo Fisher Scientific, Austin, TX, USA). Capillary electrophoresis was carried out with an ABI3130xl Genetic Analyzer (Thermo Fisher Scientific, Vilnius, Lithuania). Chromatograms were viewed and analysed using a viewer Chromas 2.6.6 (Technelysium Pty Ltd, Australia). The sequences were aligned with the reference sequence of the *SLC9A6* gene (NCBI: NM_001042537.1).

### 2.6. In Silico Analysis

In silico, Mutation Taster [15], Human Splicing Finder [16], and CRYP-SKIP [17] tools were used for predicting splice site alterations. Possible change in the amino acid sequence of SLC9A6 (UniProtKB: Q92581) protein was predicted using tools supplied in ExPASy Bioinformatics Resource Portal [18], Pfam 32.0 database [19], and UniProt database [20].

Possible structural templates for modelling SLC9A6 protein were identified with HHpred server [21]. Oligomeric states of homologous proteins were checked in the PPI3D server [22]. Further, structural models were generated by the Robetta server, using both comparative modelling (RosettaCM) [23] and contacts prediction (TrRosetta) [24] methods. The truncated version of the protein was modelled also by Rosetta ab initio protocol [25]. The quality of structural models was evaluated using VoroMQA [26]. The disordered regions of SLC9A6 were predicted by DISOPRED and MetaDisorder servers [27,28].

The study was approved by the Vilnius Regional Biomedical Research Ethics Committee. Written informed consents for genetic investigations and publication were obtained from the family.

## 3. Results

Whole exome sequencing of a tetrad (Patient 1, Patient 2, and their mother and father) identified several variants for further investigation: a splice site variant in the *SLC9A6* gene NC_000023.11(NM_001042537.1):c.899 + 1G > A, a missense variant in the gene *TAF9B* NM_015975.4:c.510T > A and another missense variant in the gene *TENM1* NM_001163278.1:c.3914G > A. Segregation analysis using the Sanger sequencing method was performed in the DNA samples of the probands, their healthy brother, and their mother. The *TAF9B* gene missense variant was not confirmed in any of the DNA samples. This could be explained by a very low mapping quality score of the DNA fragments of NGS data in this position due to the presence of several similar DNA sequences to this position in the genome. The *TENM1* gene missense variant was confirmed in a hemizygous state in the DNA samples of the healthy brother and in a heterozygous state in their mother, which proves that the variant is irrelevant for the formation of the phenotype of the affected individuals. A donor splice site variant in *SLC9A6* was confirmed in a hemizygous state in the DNA of the affected siblings and in a heterozygous state in the DNA of their mother (Figure 1C). With the use of bioinformatic tools, this donor splice site variant was predicted as likely pathogenic and pathogenic, possibly disrupting the normal splicing process. This variant is not observed in the gnomAD v2.1.1 dataset.

To elucidate the consequences of the splice site variant on messenger RNA (mRNA) structure, extraction of mRNA was performed and a cDNA sample of proband 1 was analysed by Sanger sequencing using forward primer created to hybridise on exon 5 and reverse primer to hybridise on exon 8. The sequencing results showed a single transcript composed of exons 5, 7, and 8, but lacking exon 6 (Figure 1D). In silico, the deletion of exon 6 leads to a frameshift and formation of a premature stop codon NP_001036002.1:p.(Val264AlafsTer3) (Figure 1E).

A possible impact on protein of this exon skipping is a truncated and functionally disturbed SLC9A6. To explore the protein structure in more detail, possible dimeric bacterial templates ranging from 70 to 540 residues were identified for the SLC9A6 region. The structure of this region was also predicted with high confidence both by comparative modelling and contacts prediction. These methods produced highly similar structural models for this region, in contrast to the N- and C-terminal parts of the protein. On the other hand, the N- and C-termini were predicted to most likely be disordered (Figure 2).

This explains the lack of consensus and low-quality scores for the N-terminal and C-terminal parts of the models. After analysing the structural models, we can speculate that the truncated protein is unlikely to form any structure that could form functionally relevant dimers in the membrane (Figure 3).

The pathogenicity of the identified splice site variant in the *SLC9A6* gene was assessed according to American College of Human Genetics and Genomics guidelines [14] and is considered pathogenic.

## 4. Discussion

In our study, c.899 + 1G > A, a donor splice site variant on the *SLC9A6* gene, which is located on the X chromosome, was identified in the DNA samples of two affected brothers whose phenotypes were consistent with CS. Sequencing analysis revealed their mother as a carrier of this intronic variant. Based on the predictions of several bioinformatic tools, the c.899 + 1G > A variant was postulated to disrupt the normal splicing process and presumably cause exon skipping. The most likely outcomes of alterations at acceptor (3′) or donor (5′) splice sites are exon skipping or cryptic splice site activation causing insertions, deletions, or intron retention [29,30,31]. Studies have demonstrated that an altered donor site could lead to skipping of the exon upstream the genetic change [32,33,34] as well as utilising the cryptic splice site upstream or downstream the canonical splicing site [35,36,37]. Moreover, a recent study by X. Lv et al. revealed and further confirmed that a single variant in the canonical splicing site could induce production of several transcripts of different lengths and exonic/intronic structures. In this study, two variants were identified at the same donor splice site position (c.2917 + 1) in the *COL4A5* gene. Although molecular analysis revealed three different transcripts caused by both donor splice site mutations, a difference was detected in the quantity of these transcripts: the c.2917 + 1G > A variant mainly led to the use of the cryptic splice site mutation and deletion of 96 bp in exon 33, while the c.2917 + 1G > C variant primarily caused the formation of transcript with exon 33 skipped [38]. These studies reflect on the complexity of the splicing process, which is known to depend on various *cis* (canonical splicing sites, splicing regulatory sequence elements) and *trans* (protein factors) elements including other characteristics (pre-mRNA secondary structure, extracellular signalling, order of intron removal, etc.) [39,40,41,42]. Therefore, the underlying reason for the execution of one of the consequences caused by the altered donor or acceptor splice site also depends on various factors and needs to be investigated.

To confirm the computational assumptions of normal *SLC9A6* splicing disruption, Sanger sequencing of proband 1’s cDNA sample was applied as a sufficient molecular approach to elucidate the effect of the changed donor splice site on pre-mRNA structure. The molecular results coincided with predictions: a single transcript lacking exon 6 was sequenced. The same exon 6 skipping effect was detected by Zhang et al. (2020), who investigated the positionally close splice site variant c.899 + 3_899 + 6del in *SLC9A6* [43]. According to the reviewed literature, exon skipping is one of the most probable events when the donor site is changed, and our case further confirms this observation. In silico, skipping of exon 6 leads to a frameshift and a premature stop codon formation NP_001036002.1:p.(Val264AlafsTer3). The loss of the SLC9A6 protein caused by the donor site variant identified in our patients involves a substantial part of the N-terminal transmembrane domain. Takahashi et al. (2011) showed that a frameshift mutation in the N-terminus of SLC9A6 (c.441delG, p.S147fs) causing a premature stop codon fully disrupts protein function and reduces protein quantity [44]. After mRNA analysis, researchers explained the decreased number of *SLC9A6* molecules by a naturally occurring cell process called nonsense-mediated mRNA decay (mRNA-NMD) [44]. Studies of different variants causing premature stop codons showed that aberrant mRNAs trigger the NMD process when the stop codon resides 50–55 nucleotides upstream of the last exon–exon junction. Otherwise, the altered mRNA is translated into protein, which has the potential to increase the severity of disease [45]. In our case, the premature stop codon is far more than 55 nucleotides upstream the last two exons, and therefore NMD is most probably stimulated, leading to reduction of *SLC9A6* mRNA. After consideration of the Takahashi et al. (2011) example [44], other scientific literature, and computational analysis, our hypothesis is that c.899 + 1G > A causes not only the formation of non-functional and truncated protein, but also the degradation and subsequently reduced translation of altered mRNA.

In our case, the altered donor splice site leads to the SLC9A6 protein losing more than half of its amino acids. To see if any structure could be formed at all, the sequence of the truncated protein was submitted to the Robetta ab initio modelling server, but the resulting models had no consensus and were thus probably all wrong. According to the results of structural modelling, we suggest that the truncated protein would not carry out any function and would probably be degraded. Our hypothesis is consistent with previous experimental results. Roxrud et al. (2009) observed that a small deletion in *SLC9A6* caused protein degradation [46], and Takahashi et al. (2011) found no *SLC9A6* expression in a similar case of premature stop codon [44]. To understand the disturbed molecular processes caused by *SLC9A6* truncating variants that lead to CS phenotype development, the functional structure and role of SLC9A6 protein at the cellular level needs to be explored. The main SLC9A6 cell-wise localisation is early and recycling endosomes, while transiently it is found in the plasma membrane. The main localisation reflects its specific role: SLC9A6 is involved in endosomal pH homeostasis, where an acidic environment is required, and therefore in the processing of intracellular cargo, since the endosome itself is the sorting organelle. Optimal organelle pH is maintained by translocating intracellular sodium (Na^+^) or potassium (K^+^) into the endosome in exchange for luminal hydrogen (H+) removal. The organelle-targeting and pH homeostasis function is ensured by the N-terminus, which is the larger and membrane spanning part of SLC9A6, while proper protein-protein interaction depends on the cytoplasmic C-terminal tail where important phosphorylation sites occur [47,48,49,50,51]. The N-terminal transmembrane domain consists of twelve membrane-spanning α-helices (amino acids 24–534). According to the literature, the part between fourth and tenth transmembrane helices is necessary for transporting ions [49,51,52,53]. Disturbing the function of SLC9A6 leads to hydrogen retention in endosomal lumens and ultimately to overacidification [8].

To further understand the impact of these cellular disturbances and their link to phenotypic, mainly neurological, symptoms, experiments were conducted in animal models and neuronal cells where *SLC9A6* is abundantly expressed. The studies showed SLC9A6 connection and participation in neurodevelopment, neurodegeneration, and functionality of synapses. When cells were modified by deleting the SLC9A6 coding gene, the increased acidic environment in endosomes was linked to altered signalling via the BDNF/TrkB pathway (brain-derived neurotrophic factor and its receptor, tropomycin receptor kinase B) [54]. This cellular signalling is relevant for dendritic and axonal arborisation and growth. Acidified endosomes led to early TrkB degradation and eventually to disturbed neuronal development, which can influence the manifestation of microcephaly and intellectual disability [54]. Another possible mechanism for development of intellectual disability is through AMPA (α-amino-3-hydroxyl-5-methyl-4-isoxazole-propionate) receptor trafficking by endosomes. Decreased pH in organelles caused by SLC9A6 dysfunction could alter the long-term potentiation process where AMPA receptors are essential, and therefore synaptic plasticity could be disrupted [55,56]. Neurodegeneration is related to SLC9A6 via amyloid precursor protein (APP) processing, which is dysregulated when the pH of endosomes is unbalanced in the cell culture model of Alzheimer’s disease [56,57]. Interestingly, M. Kerner-Rossi et al. (2019) studied the influence of SLC9A6 dysfunction on the sensory impairments of CS patients, particularly disrupted sensitivity to pain [56]. Scientists demonstrated that *SLC9A6* knock-out mice are less responsive to harmful thermal and mechanical stimuli. This could be explained by several cellular processes: increased GM2 ganglioside quantity in the superficial dorsal horn, which is involved in the transmission and processing of sensory information, and abnormal vesicular organelle distribution within perikarya [56]. All these functional studies are revealing the underlying processes in the pathogenesis of CS. They also highlight the need for further comprehensive studies of the most affected areas detected in CS patients. Functional assays are also necessary to reliably tackle the question of the pathogenicity of variants in *SLC9A6* and contribute to the exploration of disrupted cellular processes leading to CS.

## 5. Conclusions

To conclude, our study provides insight into the disrupted splicing process caused by donor site variant c.899 + 1G > A in the *SLC9A6* gene. Molecular analysis of the affected individual’s cDNA identified the exon 6 skipping event that leads to a premature stop codon, resulting in structurally and functionally disrupted protein, and probably triggers the nonsense-mediated mRNA decay process. These molecular changes cause the development of Christianson syndrome. Thus, the diagnosis of our patients was confirmed, allowing doctors to effectively tailor a clinical management strategy. Our study emphasises the importance of investigations of splice site variants to validate their pathogenicity. Lastly, the molecular analysis of splicing variants is fundamentally and clinically relevant to explaining the pathobiology and aetiology of various medical conditions, including neurodevelopmental disorders such as Christianson syndrome.

## Figures and Tables

**Figure 1 medicina-58-00351-f001:**
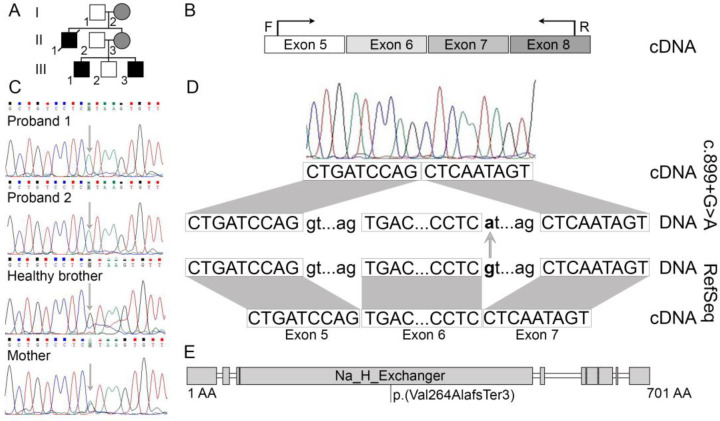
(**A**)—the genealogy of the family presenting two affected brothers in the third generation and affected uncle in the second generation, as well as female carriers in the second and first generations. (**B**)—schematic representation of the primers designed to amplificate specific cDNA fragment of the SLC9A6 gene. (**C**)—Sanger sequencing of DNA samples confirms the SLC9A6 splice site variant c.899 + 1G > A in proband 1, proband 2, and their mother. (**D**)—schematic representation of cDNA Sanger sequencing results showing exon 6 skipping. (**E**)—SLC9A6 protein linear representation with predicted impact of exon 6 skipping on protein leading to frameshift and premature stop codon.

**Figure 2 medicina-58-00351-f002:**
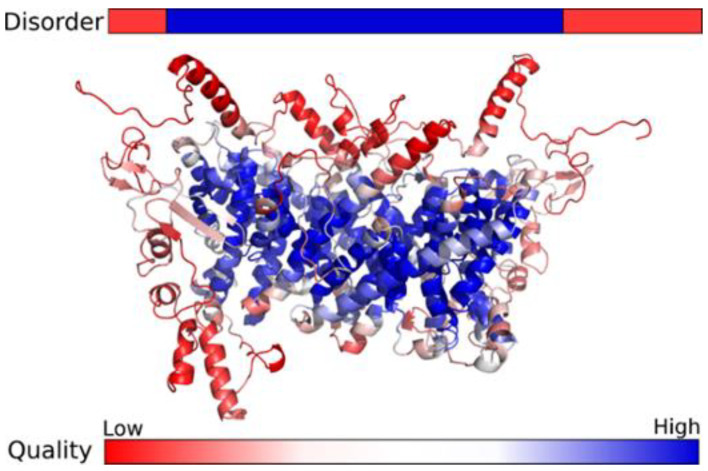
Disorder predictions and structure model of SLC9A6 dimer. Low quality regions in the structural model (red) usually correspond to parts of the protein predicted to be disordered, and the structured core of the dimer is predicted with higher confidence (blue).

**Figure 3 medicina-58-00351-f003:**
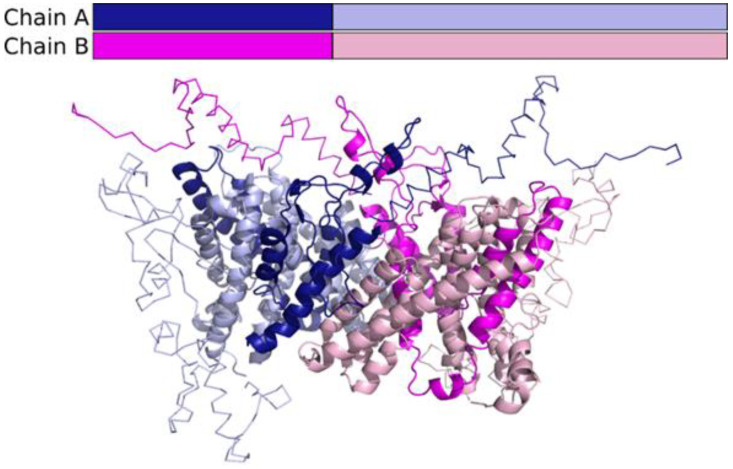
Regions corresponding to the truncated mutant protein (dark colours) in the context of the structure model of the SCL9A6 dimer. Regions predicted as disordered are shown in ribbon representation, and the structured part of the protein is shown in cartoon.

## Data Availability

All data generated or analysed during this study are included in this published article. Any additional information is available from the authors upon request.

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
