# Peer review of "Donor Splice Site Variant in *SLC9A6* Causes Christianson Syndrome in a Lithuanian Family: A Case Report"

_medicina, 2022, doi:10.3390/medicina58030351_

Round 1
Reviewer 1 Report
The authors reported a donor splice site variant in SLC9A6 in a Lithuanian family with Christianson syndrome in this case report. To explain this splicing variant relevant to the pathobiology of Christianson syndrome, the authors also present in silico analysis of this donor splice site variant in SLC9A6. This study provides insight into the disrupted splicing process caused by c.899+1G>A in the SLC9A6 gene by bioinformatical and molecular analysis. However, the relation between this variant and the pathobiology of Christianson syndrome needs to be clarified.
There is a primary concern in the current manuscript.
The variant (NM_001042537.1:c.899+1G>A) has been reported in the SNP database (rs1556617455). I wonder to know the allele frequency in the public database.
In the results, the presentation of WSE was not clear. The author used segregation analysis in the DNA samples of the probands, their healthy brother, and their mother. How about their father? Did their father have the mutant NM_001042537.1:c.899+1G>A?
How about the consequences of this splice site variant on mRNA structure from their mother, father, proband 2, and normal controls?
To get the entire figure, the authors should elucidate the effect of the changed donor splice site on pre-mRNA structure from other subjects.
Minor:
English grammar and style should be improved. I recommend that authors have the manuscript professionally edited or read by a genetic field expert.
For example,
Title: “Donor splice site variant in SLC9A6 causes Christianson syndrome in Lithuanian family” should be “Donor splice site variant in SLC9A6 causes Christianson syndrome in a Lithuanian family”
Line 18: “the pathogenic alterations of SLC9A6 is a known cause” should be “ the pathogenic alterations of SLC9A6 are a known cause”
Line 22: “Next generation sequencing” should be “Next-generation sequencing”
Line 63: “detectable in all the nervous system” should be “detectable in all the nervous systems”
Figure 1C: a typo “Healthy broher” should be “Healthy brother”.
Line 242: “transcripts of different length and exonic/intronic structure” should be “transcripts of different lengths and exonic/intronic structures”
Author Response
Dear the Editorial Board and Reviewers,
Please find enclosed a revised version of our manuscript entitled “Donor splice site variant in SLC9A6 causes Christianson syndrome in a Lithuanian family: a case report”, for consideration for publication in MEDICINA.
We have modified the manuscript according to comments, which were suggested by Reviewers. All changes to the manuscript are indicated in the text by using track changes. The answers to Reviewers’ reports together with any other comments you will find below.
We hope that you will find this revised version of our study suitable for publication in MEDICINA.
Many thanks for your expert editorial assistance.
We are looking forward to hearing from you at your earliest convenience.
Sincerely,
Gunda Petraitytė
Reviewer reports:
Reviewer 1:
Point 1: The authors reported a donor splice site variant in SLC9A6 in a Lithuanian family with Christianson syndrome in this case report. To explain this splicing variant relevant to the pathobiology of Christianson syndrome, the authors also present in silico analysis of this donor splice site variant in SLC9A6. This study provides insight into the disrupted splicing process caused by c.899+1G>A in the SLC9A6 gene by bioinformatical and molecular analysis. However, the relation between this variant and the pathobiology of Christianson syndrome needs to be clarified.
Response 1: We would like to thank Reviewer 1 for a review, comments, and questions. The relation between c.899+1G>A splice site variant and the pathobiology of Christianson syndrome is described in section “Results” and “Discussion”. We showed that splice site variant causes exon 6 skipping. In silico, deletion of exon 6 leads to a frameshift and formation of a premature stop codon NP_001036002.1:p.(Val264AlafsTer3). A possible impact on protein of this exon skipping is a truncated and functionally disturbed SLC9A6. The previous functional studies of SLC9A6 reviewed in our manuscript revealed the underlying processes in the pathogenesis of CS. Therefore, in our study we confirm the pathogenicity of c.899+1G>A splice site variant.
Point 2: There is a primary concern in the current manuscript. The variant (NM_001042537.1:c.899+1G>A) has been reported in the SNP database (rs1556617455). I wonder to know the allele frequency in the public database.
Response 2: We are thankful for the Reviewer’s 1 observant comment. The c.899+1G>A donor splice site variant was not found in gnomAD, 1000G and ExAC databases, this variant has no entry in them. Thus, the identified splice site variant is very rare according to the public databases. We added this information in “Results” section.
Point 3: In the results, the presentation of WES was not clear. The author used segregation analysis in the DNA samples of the probands, their healthy brother, and their mother. How about their father? Did their father have the mutant NM_001042537.1:c.899+1G>A?
Response 3: We appreciate Reviewer’s 1 concern. The analysis of the father’s DNA is not particularly necessary as the diagnosed condition of the probands’ is inherited in an X-linked manner. The boys inherited Y chromosome from the father and their X chromosome is maternal.
Point 4: How about the consequences of this splice site variant on mRNA structure from their mother, father, proband 2, and normal controls?
To get the entire figure, the authors should elucidate the effect of the changed donor splice site on pre-mRNA structure from other subjects.
Response 4: We appreciate Reviewer’s 1 concerns and thorough evaluation. After identifying the donor splice site variant, we were not able to reach and invite the other family members for a repeated blood collecting procedure to obtain the whole blood RNA samples. Although just in theory, the most likely outcome of analysing the other family members is that healthy individuals (the farther and the healthy brother) that do not have this variant would not have any effect, and the other proband and the mother (carrying the variant in a heterozygous state) would have the same effect on mRNA as the analysed affected individual.
Point 5: Minor:
English grammar and style should be improved. I recommend that authors have the manuscript professionally edited or read by a genetic field expert.
For example,
Title: “Donor splice site variant in SLC9A6 causes Christianson syndrome in Lithuanian family” should be “Donor splice site variant in SLC9A6 causes Christianson syndrome in a Lithuanian family”
Line 18: “the pathogenic alterations of SLC9A6 is a known cause” should be “ the pathogenic alterations of SLC9A6 are a known cause”
Line 22: “Next generation sequencing” should be “Next-generation sequencing”
Line 63: “detectable in all the nervous system” should be “detectable in all the nervous systems”
Figure 1C: a typo “Healthy broher” should be “Healthy brother”.
Line 242: “transcripts of different length and exonic/intronic structure” should be “transcripts of different lengths and exonic/intronic structures”
Response 5: We are truly thankful for Reviewer’s 1 careful revision. The manuscript has been corrected by the Expert of English language before the submission. However, we would like to apologize for the redundant spaces, grammar, and other inattentive mistakes. The text has been corrected according to all Reviewer’s 1 comments. Also, the text of the manuscript was repeatedly revised and corrected.
Reviewer 2 Report
- The title should mention the article type. (Case report)
- Abstract should provide more details about the two siblings being reported. It is advised to remove unrelated or already well-known information present in the ‘‘Background and Objectives’’ section.
- ‘‘The phenotype of CS overlaps somewhat with Angelman syndrome (MIM# 105830) and was previously referred to as X-linked Angelman syndrome’’ Please provide a small description about what differentiates these two disorders.
- What is the age of patient 3 (mother)?
- Could the authors provide images of the EEGs performed as supplementary material?
- Could the authors provide a table highlighting the already reported clinical associations with this abnormal gene? Labels: abnormal and clinical findings.
- Any of the individuals being reported had a dysmorphic feature?
Author Response
Dear the Editorial Board and Reviewers,
Please find enclosed a revised version of our manuscript entitled “Donor splice site variant in SLC9A6 causes Christianson syndrome in a Lithuanian family: a case report”, for consideration for publication in MEDICINA.
We have modified the manuscript according to comments, which were suggested by Reviewers. All changes to the manuscript are indicated in the text by using track changes. The answers to Reviewers’ reports together with any other comments you will find below.
We hope that you will find this revised version of our study suitable for publication in MEDICINA.
Many thanks for your expert editorial assistance.
We are looking forward to hearing from you at your earliest convenience.
Sincerely,
Gunda Petraitytė
Reviewer reports:
Reviewer 2:
Point 1: The title should mention the article type. (Case report)
Response 1: We would like to thank Reviewer 2 for a review, comments, and questions. We have modified the title, i.e., “case report” has been included.
Point 2: Abstract should provide more details about the two siblings being reported. It is advised to remove unrelated or already well-known information present in the ‘‘Background and Objectives’’ section.
Response 2: We are thankful for Reviewer’s 2 observant suggestion. We have taken this comment into account and corrected the abstract according to the suggestion.
Point 3: ‘‘The phenotype of CS overlaps somewhat with Angelman syndrome (MIM# 105830) and was previously referred to as X-linked Angelman syndrome’’ Please provide a small description about what differentiates these two disorders.
Response 3: We appreciate Reviewer’s 2 thorough analysis. The differential characteristics between Christianson syndrome and Angelman syndrome have been reported by Liu et al. (2022)[1]. CS follows an X-linked recessive inheritance pattern, with clinical features that overlap with those of AS in male patients presenting with moderate to severe global developmental delay, epilepsy, absent or impaired speech, truncal ataxia, ophthalmoplegia, acquired microcephaly, hyperkinesis, cerebellar atrophy and reduced life expectancy. The main difference among these two syndromes is distinct genetic cause of the disease – CS is X-linked recessively inherited disorder, while Angelman syndrome is caused by homozygous variant in the UBE3A. Moreover, CS can be specifically characterized by possible developmental regression, progressive cerebellar atrophy, electrical status epilepticus during sleep, and shorter life span.
We have taken this comment into account and included short description into the text.
Point 4: What is the age of patient 3 (mother)?
Response 4: We are thankful for Reviewer’s 2 question. The patient 3 (mother) is a 40-year-old. We have added this information into the text.
Point 5: Could the authors provide images of the EEGs performed as supplementary material?
Response 5: We appreciate Reviewer’s 2 request. Unfortunately, the images of the EEGs are not available, and we are not able to provide them.
Point 6: Could the authors provide a table highlighting the already reported clinical associations with this abnormal gene? Labels: abnormal and clinical findings.
Response 6: We appreciate Reviewer’s 2 suggestion. The clinal associations with different SLC9A6 pathogenic variants have been most recently reported by Zhang et al. (2022)[2], therefore such analysis would duplicate already published information and wouldn’t add any additional value. Moreover, our study mainly aims to evaluate the molecular consequences of a novel splicing variant at the mRNA level.
Point 7: Any of the individuals being reported had a dysmorphic feature?
Response 7: We are thankful for Reviewer’s 2 observant question. The reported individuals have no dysmorphic features.
[1] Xiaorui Liu, Lingling Xie, Zhixu Fang, Li Jiang. Case Report: Novel SLC9A6 Splicing Variant in a Chinese Boy With Christianson Syndrome With Electrical Status Epilepticus During Sleep. Front Neurol. 2022 Jan 14;12:796283. doi: 10.3389/fneur.2021.796283. eCollection 2021.
[2] Xiaoge Zhang, Xiaofang Wu, Hongli Liu et al. Christianson syndrome: A novel splicing variant of SLC9A6 causes exon skipping in a Chinese boy and a literature review. J Clin Lab Anal. 2022 Jan;36(1):e24123. doi: 10.1002/jcla.24123. Epub 2021 Nov 17.
Round 2
Reviewer 1 Report
Thank you for addressing my comments. The revised manuscript has been sufficiently improved for publication.